# Intelligent Bi-LSTM with Architecture Optimization for Heart Disease Prediction in WBAN through Optimal Channel Selection and Feature Selection

**DOI:** 10.3390/biomedicines11041167

**Published:** 2023-04-13

**Authors:** Muthu Ganesh Veerabaku, Janakiraman Nithiyanantham, Shabana Urooj, Abdul Quadir Md, Arun Kumar Sivaraman, Kong Fah Tee

**Affiliations:** 1Department of Electronics and Communication Engineering, K.L.N. College of Engineering, Pottapalayam 630612, India; 2Department of Electrical Engineering, College of Engineering, Princess Nourah bint Abdulrahman University, P.O. Box 84428, Riyadh 11671, Saudi Arabia; 3School of Computer Science and Engineering, Vellore Institute of Technology, Chennai 600127, India; 4Digital Engineering Services, Photon Inc., DLF Cyber City, Chennai 600089, India; 5Department of Civil and Environmental Engineering, King Fahd University of Petroleum and Minerals, Dhahran 31261, Saudi Arabia

**Keywords:** heart disease prediction, wireless body area network, optimal channel selection, improved dingo optimizer, one dimensional-convolutional neural network, modified bidirectional long short-term memory, autoencoder

## Abstract

Wireless Body Area Network (WBAN) is a trending technology of Wireless Sensor Networks (WSN) to enhance the healthcare system. This system is developed to monitor individuals by observing their physical signals to offer physical activity status as a wearable low-cost system that is considered an unremarkable solution for continuous monitoring of cardiovascular health. Various studies have discussed the uses of WBAN in Personal Health Monitoring systems (PHM) based on real-world health monitoring models. The major goal of WBAN is to offer early and fast analysis of the individuals but it is not able to attain its potential by utilizing conventional expert systems and data mining. Multiple kinds of research are performed in WBAN based on routing, security, energy efficiency, etc. This paper suggests a new heart disease prediction under WBAN. Initially, the standard patient data regarding heart diseases are gathered from benchmark datasets using WBAN. Then, the channel selections for data transmission are carried out through the Improved Dingo Optimizer (IDOX) algorithm using a multi-objective function. Through the selected channel, the data are transmitted for the deep feature extraction process using One Dimensional-Convolutional Neural Networks (ID-CNN) and Autoencoder. Then, the optimal feature selections are done through the IDOX algorithm for getting more suitable features. Finally, the IDOX-based heart disease prediction is done by Modified Bidirectional Long Short-Term Memory (M-BiLSTM), where the hyperparameters of BiLSTM are tuned using the IDOX algorithm. Thus, the empirical outcomes of the given offered method show that it accurately categorizes a patient’s health status founded on abnormal vital signs that is useful for providing the proper medical care to the patients.

## 1. Introduction

Heart disease is said to be a group of multiple diseases that create a huge impact on the veins in the human heart. Heart disease identification and analysis are performed by highly experienced doctors [1]. Different factors take a major part in generating heart disease are diabetes, junk foods, smoking, diet, age, being overweight, etc. In recent years, most hospital management created software to monitor patient records. These types of procedures became popular among the patient and generated a huge amount of patient information [2]. These types of data are rarely utilized for attaining medical-related decisions [3]. Medical application holds an enormous amount of unused data with them and they produce high complexity to convert it into effective information and so, professional support is needed to attain an effective decision [4]. All these factors are subject to research in the field of medical image processing because only a limited number of professionals are available, which makes a lot of cases to be examined wrongly so, and therefore, an efficient automation model is highly essential [5]. The main scope of the automated method is to categorize the key factor of medical data with the help of classifiers to perform effective cardiac disease analysis in the early stage. 

Various enhancements in Integrated Circuits (IC), wireless technology, physiological parameters, and electronic devices are utilized to generate a short-range communication network to monitor the patient health status in WBAN, and it is required with more data for performing effective health care monitoring [6]. Continuous power interruption may lead to life-or-death conditions and unfavorable situations. Thus, energy consumption plays an essential role in the WBAN system. Battery capacities of Sensor Node (SN) are comparatively tiny because of node size [7]. Most of the WBAN energy efficiency-based research is mainly focused on the transmission between the gateway and SN [8]. As contrasted with SN, the gateway node is required with high battery capacity to execute multiple heavy tasks like computation, long-range wireless communications, and display and so, energy consumption challenges to the gateway node [9]. Remote health monitoring is allowed to permit WBAN for accepting the technology and achieved physiological signal-delivering and acquiring processes in the monitoring system [10]. The basic WBAN model holds multiple SN and gateway nodes. SN can sense sending data and physiological signals in the gateway node with the help of short-range communication technology [11]. However, the gateway nodes are connected to a Wi-Fi network to forward the acquired data to the medical center, the Wi-Fi network suffered from limited coverage area and cannot offer effective e-health services [12]. Wireless body sensors use transducers to perform transceiver circuitry, signal detection, and power source for wireless linkages [13]. WBAN mostly utilized battery-powered wireless biological sensors for transmitting and measuring vital information with remote health units [14]. Since small-sized biological sensors in WBAN devices are unable to replace more easily since the durability of WBAN is high based on their size and the other hand, the tiny-sized battery affects the operational lifespan of WBAN.

Deep learning as well as machine learning plays an essential role in decision processing with a huge amount of information [15]. The traditional data investigation model consists of clustering and a neural network as the classification model to perform effective analysis. Data generation can be performed in multiple ways with a particular data type and they are important for developing a new technique to deal with information qualities [16]. In the Internet of Things (IoT), a superior number of advantages are utilized to create progressive information without creating any problems. Cloud, gateway and physiological sensors are the basic building blocks of a health monitoring unit [17]. Heart rate displays the heart sound and offered guidance in regulating the cardiovascular system. A few of the heart disease and treatment-related recent existing works are discussed below. In [18], has validated and explored the prospective strategies of Dan-Shen Decoction in treating Ischemic Heart Disease (IHD) by fusing the evaluation of network pharmacology. In [19], have to study the predictors and prevalence of depressed patients with coronary heart disease (CHD). In [20], have implemented prediction approaches for the risk of Coronary Heart Disease (CHD) and heart failure (HF). In clinics, heart rate is regulated under multiple controlled conditions like heart sound, blood, and electrocardiogram (ECG), but it cannot be monitored in a home environment. Various complexities achieved in the baseline method lead to the initiation of a novel heart disease prediction model for securing an efficient disease detection rate. 

Some of the essential contributions included in the offered model are listed here. 

To structure a heart disease prediction method in WBAN using the heuristic model and deep learning approaches to predict heart disease at the primary period in heart disease-affected individuals.To select optimal features by selecting essential features of heart disease using suggested IDOX for efficient heart disease prediction.To design an improved model named MBiLSTM for the forecasting of heart disease by tuning the BiLSTM parameter with developed IDOX for enhancing the prediction accuracy.To integrate an enhanced heuristic model named IDOX for selecting the significant features and also to optimize the number of suitably hidden neuron counts in BiLSTM to enlarge the accuracy of heart disease prediction.To evaluate the effectiveness of the offered IDOX-based heart disease prediction method in WBAN with multiple baseline deep structured architectures and algorithms.

The residual phases of this task are provided here. Literature works based on existing heart disease prediction in WBAN are explained in Section 2. Data collection is discussed in Section 3. Optimal channel selections in WBAN with an improved optimization approach are presented in Section 4. Modified BiLSTM-based heart disease predictions in WBAN are elaborated in Section 5. Multiple outcomes and discussions executed on the proposed model are clarified in Section 6 and finally, Section 7 has completed the given task. 

## 2. Literature Work

### 2.1. Related Works

Existing heart disease prediction model in WBAN with deep learning approaches:

In recent years, researchers [1] have developed an improved linear model-based recursion-enhanced Random Forest (RF) for detecting cardiac disorders. The developed model identified an essential feature to identify heart disease by expert systems. In the developed model, essential factors utilized for the analysis of heart disease were studied. Essential variable compression was displayed with the “Internet of Medical Things (IoMT)” paradigm to perform data investigation. Different analyses were developed to perform an effective detection rate concerning the F-measure ratio, accuracy, and stability ratio. Scientists [2] have initiated a controller STM32 to observe valvular heart disease and the developed framework was fused with deep learning approaches to enhance the fitting range. Different analysis was suggested to observe the blood flow by blocking and releasing it to analyze the changes attained in the fingertip at surface temperature. Eighteen subjects were utilized for the analysis, and one subject was utilized to observe the arrhythmia and cardiac valve insufficiency. The developed valvular heart disease identification model effectively analyzed the disorder by utilizing signals. 

Researchers [3] have recommended an end-to-end Convolution and Recurrent Neural Network (CRNN) framework to perform automated detection for five different modules of cardiac auscultation with the help of raw phonocardiogram (PCG) signal. The suggested automatic models included two different learning stages that were sequence residual learning and representation learning. Three Convolutional Neural Network (CNN) parallel pathways were implemented in the training stage for learning the fine and grained attributes of PCG. In the learning stage, the network can acquire effective time-invariant attributes and converge with superior speed. The occurred stimulation outcome displayed that the developed end-to-end framework attained effectively superior outcomes to conventional models. Researchers [4] have initiated the Time–Frequency-Domain (TFD) with the deep structure to perform automated identification of Heart Valve Disorders (HVDs) by utilizing PCG signals. The Deep CNN classifier was utilized to identify four HVD models with the help of TF images in the PCG signals by utilizing baseline transformation approaches. The suggested model was validated by utilizing the PCG signal in the public dataset. The suggested model achieved an effective heart disease detection rate than the existing approach using PCG signal.

Researchers [7] have initiated a joint scheduling and admission control model to tune the energy efficacy in terms of intra in WBAN link. Various issues in Markov decision processes, Lagrange multipliers, and relative iteration values were considered for resolving with an intelligent optimal approach. Various validation analyses performed on the suggested approach secured an effective high detection rate than other models. Thus, the developed model attained efficient power saving in the SN when contrasted with multiple scheduling approaches. Researchers [8] have proposed a modified neural network-based deep structure monitoring approach to perform cardiac analysis and medication. The suggested model utilized three different steps that are classification, authentication, and encryption. Initially, communicated the sensor data to the cloud platform with a help of a wearable IoT sensor device which was fixed in a human body. Then, the sensor data were enciphered and safely communicated to the cloud with the help of the Prediction of Heart Diseases based-Asymmetric Encryption Standard (PHD-AES) approach. The enciphered data were deciphered and further classification was performed by utilizing Deep Learning Modified Neural Network (DLMNN) classifier. The simulation outcome of a developed model achieved an affected detection rate than the conventional approach.

B.Existing heart disease prediction model in WBAN with heuristic algorithms:

Researchers [5] have suggested a WBAN approach according to a three-tier structure to perform early detection of Congenital Heart Diseases (CHD). Media Access Control (MAC) address IEEE 802.15.6 was utilized to detect the data superiority level. Different data were scheduled in Time Division Multiple Access (TDMA) according to priority level. Then, data transmission was performed in the first tier with the help of the standard Encipher approaches. In the second tier, accurate channel selection was performed by utilizing Bird Swarm Optimization (BSO) approach. Different simulations were performed in the developed model concerning power consumption, average delay, and probability in presented channels. Researchers [6] have suggested a hybrid fuzzy-related decision tree approach which was utilized to offer an effective prediction rate in the early phase with a remote and continuous patient monitoring system. Various outcomes attained in the developed approach were contrasted over different classifiers and attained a higher accuracy rate. 

C.Existing heart disease prediction model in WBAN with Cross-Layer Design Optimal (CLDO) framework:

Researchers [21] have implemented a Cross-Layer Design Optimal (CLDO) framework to tune the parameters like a lifetime, transmission reliability, and energy efficiency of WBANs from various regions. Additionally, the author has introduced a relay decision algorithm to choose enhanced relay nodes. It has been utilized to choose the relay nodes that make sure the stability of network energy consumption. Especially, a synthesized cross-layer scheme for optimization is offered to balance the transmission power of every node in the given network. At last, the empirical findings of the designed method have confirmed its effectiveness and also it was utilized to enhance the transmission power and network lifetime. Researchers [22] have developed the first robust optimization model for routing in BAN over traffic uncertainty and tuning of network topology. It has utilized appropriate linear mitigations to conduct a randomized fixing of the constraints, and also it has been supported by an accurate large variable neighborhood search. Finally, the simulation findings have established that the designed method has provided an enriched performance.

D.Existing heart disease prediction model in WBAN with Integer Linear Programming (ILP) robust method:

Researchers [23] have implemented an actual heuristic for its solution. It integrated probabilistic and deterministic variable fixing schemes. It was conducted by the statistics coming from better quality linear relaxations of the Integer Linear Programming (ILP) robust method. Additionally, it holds a huge neighborhood search for reparation and development of constructed solutions. It has been utilized to resolve the ILP problem. The computational tests on the designed method have shown that it has found better solutions than the other conventional approaches. Researchers have implemented a cross-layer routing strategy for WBAN superiority of service development. The proposed algorithm has extended the network lifetime and also it has been utilized to reduce the node energy consumption for WBAN applications. It has developed in two phases. In the first phase, it was designing a reliable link routing policy and energy efficiency in the network layer. The second phase has been used for developing the protocol of the model. Finally, the empirical findings of the offered method have confirmed that the designed method has attained enriched performance. Researchers have presented a general idea of BAN, and also it has described BAN communication types and their limitations. It has presented a taxonomy of BAN projects based on the year of publications. The downsides and enhancements of the given network were utilized for the upcoming works. 

E.Heart disease prediction model in WBAN with recent existing works:

Researchers [22] have promoted a more robust technique that fused a grid of stacked autoencoders with deep learning strategies. The recommended stacked autoencoders were utilized to attain the robust set of features and also the neural network was categorized as the recently acquired set of features. Especially, the strong statement of this work was to validate the Medical Decision Support Systems (MDSS) with the help of publically available datasets. At last, the simulation outcomes of the designed method have revealed that it has attained elevated classification accuracy.

Researchers [24] have investigated diverse expert systems like Support Vector Machine (SVM), Naïve Bayes (NB), Decision Tree (DT), and also the 1D CNN-LSTM to implement a system that assists physicians through continuous monitoring purposes. Thus, the empirical outcomes reveled that it has yielded efficient, fully connected monitoring systems, and is cost-effective for cardiac patients.

Researchers [25] have implemented a novel stacking ensemble ML framework. This framework has acted as the best ML model to calculate relative feature weights. Here, the early risk assessment was a significant dimension that was utilized to reduce CVDs diseases. Moreover, deep structured architectures like ANN and SVM were transformed into heart risk scores and simple statistical models. At last, the simulation outcomes of the designed method have revealed that it has attained elevated performance.

### 2.2. Problem Statement

Various sensors utilized to monitor the body parameter of heart patients are referred to as body nodes with different power consumption rates. Major risk factors attained in the nodes are mobility, communications, security, and energy, where security is considered the major issue that needs to be resolved. The existing approaches utilized for heart disease detection with deep learning approaches are showcased in Table 1. RFRF-ILM [1] is used to combine random features as well as linear features to offer an accurate disease prediction rate, and also it effectively minimized the time and cost of the system. However, it needs a superior amount of data to perform effective analysis. STM32 [2] enhanced the analysis and curve fitting rate rapidly when the input data is provided. However, it consumes more time and the system is cost-effective. CRNN [3] offered a high efficacy rate and a non-latency rate. At the same time, it can use only limited data for the analysis. TFDDL [4] proved superior clarity and maintain simplicity at the time of system analysis. However, it required a transfer function to perform the simulation, and they lack in time and frequency resolution. Three-tier network [5] offered a highly efficient outcome with an enhanced convergence rate by utilizing multiple parameters. Moreover, it consumed more time to resolve the occurred issues. A hybrid Fuzzy Decision Tree (HFDT) [6] effectively removes the redundant structure attained in data and provides an enhanced detection rate. However, it required a huge amount of time to connect, and they are highly sensitive to the noisy background. Markov decision processes [7] tune the energy efficacy attained in the WBAN node as well as the gateway node and also enhanced the throughput. Further, it did not consider the power consumption and so, the system moved towards high complexity. DLMNN [8] utilized a highly secured data transfer process for achieving an optimal outcome. Nevertheless, it did not have the efficacy to monitor heart disease in real-time. So, it is essential to structure an advanced system for heart disease detection in WBAN using deep structured architectures.

### 2.3. Discussion

Thus, there is a need to develop an IDOX algorithm for selecting accurate features and also for efficient heart disease prediction. This performance improvement for the designed method applies to real-time applications like mobile, clinical, and healthcare applications. Especially, the MBiLSTM model is implemented for enhancing prediction accuracy. The tuning of constraints for deep learning strategies is done by the IDOX algorithm. It is utilized to reduce optimization and overfitting issues. The empirical outcomes of the deigned method proved that it attained better accuracy and precision rate. Table 1 shows the superiorities and downsides of existing heart disease prediction model in WBAN.

## 3. Problem Formulation and Data Collection Procedure Used in Optimal Channel Selection for Heart Disease Prediction

### 3.1. Problem Formulation

Health reports can provide information about the country’s clinical facilities accurately. In general, the death rate is enlarged according to different heart diseases such as heart attack, heart failure, etc and it leads to the structure of an effective prediction of heart disease approach in recent days. Huge numbers of prediction systems are initiated to perform remote health monitoring in the complex background along with multiple communication protocols. WBAN has multiple standards and complexity, and they are discussed as follows. IEEE 802.15.6 develops slotted-Aloha MAC protocol and holds eight superior priorities and they are elaborated over contention probability. First, the basic value is allocated for contention probability, and depending on the slot, nodes are allocated to Aloha. If the communication is failed then the value of contention probability is split in half and it produces a collision. In emergency data, high back-off time is provided to superior contention probability which provides an enhanced delay rate to transmit the emergency information. Then, the priority-based adaptive MAC (PA-MAC) protocol in WBAN is utilized to classify the incoming packet as critical, non-medical, on-demand, and normal classes. Several channels such as beacon and data channels are highly focused to avoid a collision. However, TDMA is utilized as emergency data which rapidly enhances the waiting time. The nodes include short distance along with superior residual power in a base station and they are selected as the best forwarder node and the entire node consumes more time. But, to satisfy Quality of Service (QoS) requirements, different scheduling approaches are developed in WBAN. Likewise, two-stage allocation models for a channel are designed and involved intra and inter-communication in WBAN. The channel allocation model could not consider the priority level in the data and the network efficacy also reduces rapidly. In the entire SN, data classification is offered to attain a secured and reliable data transmission rate. Game theory is utilized to remove the interference in the system but they are not feasible for immediate data communication. Thus, scheduling and data transmission become highly challenging issues in the WBAN system. So, the developed model tried to offer effective optimization outcomes in the early heart disease detection structure over heuristic approaches. 

The accessible channels are identified by the WBAN administrator by developing a cognitive radio module with spectrum sensing [26]. Channel selection is performed to analyze the stability required to perform transmission. The channel stability is provided in Equation (1).
(1)$$DS_{g}=\frac{1}{2a}\sum_{j=1}^{2a}\xi_{j}\times \xi_{j}$$

Here, the term $\xi_{j}$ indicates the Gaussian noise signal along with unit variance and zero means, and a indicates the product time bandwidth. Optimal power selection is performed by utilizing two different cases. The maximal number of successful transmissions is attained when the channel is chosen and also minimal transmission outcome is attained at the time of retransmission. Thus, to attain an optimal channel with a huge number of successful transmissions, an effective enhancement is required.

The conventional research works on WBAN are affected by different demerits such as data aggregation, inefficient channel selection, and ineffective scheduling over network congestion. The above-mentioned complexities may affect the prior prediction of heart disease in the individuals and so, optimal selection of channels in WBAN along with an accurate scheduling approach help early analysis, and they are discussed in the developed model.

### 3.2. Experimental Dataset

The offered prediction of heart disease framework based on WBAN is designed by extracting diverse types of data from 1000 individuals with three different attributes such as respiratory stage, heartbeat rate, and oxygen stage from health records. The data are acquired randomly with a precise range according to [5,27,28]. Several values that occurred in SN are utilized in WBAN over the base station and hub. In this work, heart disease prediction is done based on three attributes that are described below.

(i)Oxygen stage: If the oxygen rate is below 92 then it shows a positive result.(ii)Respiratory stage: If the respiratory rate is above 53 then it shows a positive result.(iii)Heartbeat rate: If the heartbeat rate is below 100 then it shows a positive result.

### 3.3. Developed Model

Network sensors are considered an undisturbing and comfortable way to observe the day-to-day health activity of the patient. This type of monitoring is possible with the WBAN system because it has utilized small Bio-Medical Sensors (BMS) which are cost-effective. The BMS monitoring is performed by Body Area Network Coordinator (BANC) as well as star entire network. Patient monitoring is classified as emergency data and non-immediate data to offer an effective prediction rate. The MAC design utilized IEEE 802.15.4 protocol. Different types of patient data are classified as Reliability Data Packets (RP), Delay Data Packets (DP), Critical Data Packets (CP), and Ordinary Packets (OP). This type of categorization helped to deliver the data without any loss to the doctors and the low energy consumption is attained in BMS which leads to a lack of consideration in high and low threshold values in emergency classification. These types of issues are tackled by utilizing MAC protocol and also slot allocation is done according to the alert signals in BMS. According to research, heart disease is recognized as life threading problem globally and they are not classified based on visual perspective. However, regular observations like glucose levels, heartbeat, temperature, blood pressure, respiratory rate, and oxygen level are considered useful elements to identify the disease in its early stage. Communication attained among various parts of sending data are analyzed with different techniques. Due to the availability of multiple approaches, it is a complex task to select an accurate prediction model from the conventional approaches. All of these techniques have several functional features and they need to consider various limitations when utilized in healthcare applications. A different form of heart disorder may collapse the internal organs and is subjected to cardiovascular disorder. So, an ideal prediction model is utilized to solve these issues accurately but they need more data to perform accurate heart disease analysis in individuals and it stayed as a most important research gap. Various deep learning approaches such as decision trees, CNN, and Artificial Neural Networks (ANN)are commonly utilized in disease detection approaches but they are incapable of predicting heart disease in a real-world system. So, to resolve all the complexities attained in the baseline method, it is essential to design a new framework for heart disease prediction in the WBAN system. The structural view of the suggested prediction of the heart disease method is represented in Figure 1.

The latest heart disease prediction model in WBAN is structured based on a deep learning model to attain an effective prediction rate of heart disease at the prior period in heart disease-affected individuals. Various data such as heartbeat rate, respiratory rate, and oxygen level are taken from a benchmark database and they are fed to the channel selection stage. Then, the attained channels are fed to the deep feature extraction phase, where the deep features are extracted using 1DCNN an autoencoder. Then, the acquired deep features are subjected to the concatenation phase, and further the concatenated features are given to the feature selection stage. Here, the significant features are chosen with the help of enhanced IDOX. Later the selected optimal features are subjected to the prediction phase and prediction is performed with the help of modified Bi-LSTM and its constraints such as several suitable hidden neuron counts are tuned by utilizing developed IDOX to maximize the accuracy rate for achieving a better heart disease prediction rate.

## 4. Improved Dingo Optimization for Accurate Channel Selection in WBAN

### 4.1. Improved Dingo Optimizer

The designed optimization model IDOX is used to optimize hidden neuron count in BiLSTM for providing an effective heart disease prediction rate. DOX [29] required minimal computation time and mathematical efforts to find out the best optima. Though, attaining search space in a real-world system is highly complex because it includes more local optimal solutions in search space. The group attack, scavenger, survival, and persecution are upgraded by Equations (2)–(4).
(2)$$\alpha =\left(BFst-WFst\right)$$
(3)$$X=\frac{mean\left(Dst\left(Fst-WFst\right)\right)}{mean\left(Dst\left(Fst-BFst\right)\right)}$$
(4)$$UB=\alpha \times \left(\frac{c}{X}\right)$$

Here, the terms BFst denote the best objective function and WFst indicates the worst fitness function, Fst refers to the fitness, Dst denotes the distance, X and c are the general terms, and UB is the arbitrary number in the limit $\left(0,1\right)$.

DOX is considered a biological approach to performing global optimization and it easily imitates the hunting planes of dingoes. Diverse hunting plans followed by dingoes are scavenging, grouping tactics, and persecution.

Stage 1 (Group attack): Group attack is performed when dingoes are ready to perform hunting. The dingo can detect the prey location easily and if the prey is detected, then the entire dingo starts to cover it and the dingo habit is given in Equation (5).
(5)$${\vec{M}}_{b}\left(z+1\right)=\beta_{1}\sum_{q=1}^{hi}\frac{\left[\vec{\phi s\left(m\right)}-m\vec{b}\left(z\right)\right]}{hi}-\vec{m}\times \left(z\right)$$

The novel location of the search candidate is given as $\vec{m}\left(z+1\right)$, a random integer number is given as *hi*, the best iteration attained from the existing iteration is denoted as $m\vec{b}$, the current search agent is referred to as $\vec{m}\times \left(z\right)$, and sub-set offered for all the search agents is termed as $\phi s\left(m\right)$ and uniformly produced arbitrary number presented in the limit of $\left[-2,2\right]$ is given as $\beta_{1}$ and it is updated in Equation (4).

Stage 2 (Persecution): The major scope of the dingo is to catch the tiny prey and chase them with full power until it occurred along and the dingo behavior is elaborated in Equation (6).
(6)$${\vec{m}}_{b}\left(z+1\right)=\vec{m}\times \left(z\right)+\beta_{1}\times f^{\beta_{2}}\times \left({\vec{m}}_{c_{1}}\left(z\right)-{\vec{m}}_{b}\left(z\right)\right)$$

Here, the term $\vec{m}\left(z+1\right)$ is considered a dingo movement in the current search agent ${\vec{m}}_{b}\left(z\right)$. The attained random variable in the range of $\left[-1,1\right]$ is denoted as $\beta_{2}$, a random number is given as $c_{1}$ and it is updated in Equation (4), and also ${\vec{m}}_{c_{1}}\left(z\right)$ is the search agent for cth interval.

Stage 3 (Scavenger): The scavenger features are considered as action when dingoes discover their food and this behavior is provided in Equation (7).
(7)$${\vec{m}}_{b}\left(z+1\right)=\frac{1}{2}\left[f^{\beta_{2}}\times {\vec{m}}_{c_{1}}\left(z\right)-{\left(-1\right)}^{\sigma }\times {\vec{m}}_{b}\left(z\right)\right]$$

The binary number is denoted as σ and they are generated in an unsystematic way and are updated in Equation (4).

Stage 4: The survival rate of the dingo is provided in Equation (8).
(8)$$sur\left(D\right)=\frac{BFst-Fst\left(D\right)}{BFst-WFst}$$

The best fitness is termed as BFst and the least fitness function is given as WFst in the present generation. Fitness value for Dth search candidate is offered as $Fst\left(D\right)$ and also the minimal survival rate is attained in Equation (9).
(9)$${\vec{m}}_{b}\left(z+1\right)=\vec{m}\times \left(z\right)+\frac{1}{2}\left[{\vec{m}}_{c_{1}}\left(z\right)-{\left(-1\right)}^{\sigma }\times {\vec{m}}_{c_{2}}\left(z\right)\right]$$

Here, the minimal survival rate is indicated as m→bz, chosen search agent for the random number c1 and c2 are given as m→c1z and m→c2z, respectively and these random numbers are updated in Equation (4). Pseudo code for offered IDOX is provided in Algorithm 1. The decision variables and the feasibility constraints for the designed **IDOX** are given in Table 2.
**Algorithm 1:** Proposed IDOXInput: Hidden neuron count in BiLSTM is given as HdiBilstm. Output: Optimal range of hidden neuron count.Generating algorithm BeginInitialize dingo population and parametersInitialize the current best solution based on a fitness functionFor all solutionAssign fitness for entire individualsAllocate constraints to the solutions Update group attack by Equation (5) Update persecution by Equation (6) Update scavenger by Equation (7) Update survival rate by Equation (8) Final updating takes place by Equation (4) Find the accurate solutionEnd

### 4.2. Data Aggregation

The basic phase of the offered detection of heart disease approach is data aggregation and it holds *e*_−_
number of WBAN that are referred to as neonates $P_{1},P_{2},\cdots ,P_{e}$, then the entire neonate holds h—number of SN as $G_{1},G_{2},\cdots ,G_{h}$ which is utilized to acquire biomedical data. Collision is avoided with the help of sensed data by considering TDMA time slots.

Different alterations are performed in the header format of MAC to alter the field data for IEEE 802.15.6. Various data collected by SN break the usual threshold value and they are defined as the irregular measurement and are allocated by superior priority when the transmission is performed. In sensed data, the critical level is defined by utilizing field data type. Several data models offered with MAC header hold critical as well as non-critical data with it and they are said to be binary values that are used to reduce the overhead.

Critical type binary value is denoted as 00 along with superior priority level and so, the non-critical is termed as 01 by minimal priority range. In the initial tier, h—SN is attached over a single hub and it creates multiple collisions when the transmission is performed among the data. But it is not proven that all sensors utilize the critical data similarly. Hence to lower the data transmission in collision scheduling, sensed data is highly essential. TDMA is utilized to perform scheduling all time frames FT that are split into multiple time slots *w* as provided in Equation (10).
(10)$$FT=FT_{1},FT_{2},\ldots ,FT_{w}$$

At the time the data transmission is performed, several time slots are allocated to their entire node. The SN communicates with the help of its data towards the hub which is assigned to a time slot and the developed model assigned the data over priority-related scheduling.

Scheduling is structured between the entire relative hub and neonate. However, in several WBANs, scheduling in SN is provided to the respected data type. TDMA scheduling is included to perform fully to avoid a collision at the time of transmission and it reduces the transmission time when critical data is transmitted.

Basically, medical data cannot withstand multiple attacks, and also protecting data aggregation is a highly essential task. Advanced Encryption Standard (AES) is a symmetrical approach that utilizes a similar key to perform decryption and encryption. The entire SN provided to WBAN performs encryption initially in sensed data with the help of the secret key $\kappa_{Fb_{v}}$ of vth hub $Fb_{v}$. SN node transfers sensed data to a hub at a time slot $Fb_{v}$ which is offered in Equation (11).
(11)$$DS_{n}\to {\left(E\left(DS_{n}\right)\right)}_{Fb_{v}}$$

Here, encrypted data is given as $E\left(DS_{n}\right)$ which occurred in $Fb_{v}$ hub, and sensed data attained in $Y_{k}$ is referred to as $DS_{n}$ in nth WBAN. Later, acquired data is decrypted with the help of a hub from the entire SN by utilizing a similar key $\kappa_{Fb_{v}}$ along with AES. The attained data for stimulation are indicated as $DA_{wa}^{inp}$ and they are provided as input for the channel selection phase. Here, the term wa=1,2,…,WA and the total number of attributes are denoted as WA.

### 4.3. IDOX-Based Channel Selection

The attained data $DA_{wa}^{inp}$ is provided as the key input for the channel selection stage. Data aggregation is performed according to TDMA and the optimal channel selection is executed using an enhanced heuristic technique termed IDOX. Channel selection is employed in the second tier and the accurate channels are chosen by a hub to perform the transmission of aggregated data to a base station. According to tier one, the entire SN is attached to several hubs and then in the second tier, each hub is communicated with a base station. Collision is avoided by utilizing CSMA/CA along with optimal channel selection and developed IDOX.

In the base station and hub, a group of channels $\left\{Ch_{1},Ch_{2},\cdots ,Ch_{N}\right\}$ are used to transmit the data. Hubs start to observe several channels to determine idle channels with the help of CSMA/CA. here, all the nodes get a channel and forward the data to avoid a collision before the transmission process is performed. Accordingly, entire idle channels are recognized by the hub and the accurate channel is selected from the idle channel by utilizing the IDOX approach. The acquired channel is denoted as $DA_{wa}^{CH}$ and is offered in the deep feature extraction phase.

## 5. Modified BI-LSTM-Based Prediction of Heart Disease in WBAN through Optimal Channel

### 5.1. Deep Feature Extraction Using IDCNN

The acquired data $DA_{wa}^{CH}$ from the channel selection phase is designed as the input for 1D-CNN and autoencoder in the deep feature extraction phase.

Autoencoder [27]: The acquired data $DA_{wa}^{CH}$ from the channel section is offered as the input to the extraction of deep features to attain the first set of deep features. Autoencoder and CNN have a similar structural design in a network system and also these structures have common components such as convolution filters and pooling layers. The output node and input node have common measurements and they are treated as essential features. However, in the learning process, the system provides data to the node and they are not labeled as a dependent system. The autoencoder cannot follow any specialized approach to perform effective data processing and the feature map for uth inactive representation is displayed in Equation (12).
(12)$$l^{u}=\sigma \left(t\times W^{u}+k^{u}\right)$$

Mono-channel input is denoted as *t*, the activation function is termed as *σ*, 2-D convolution is displayed as ∗, and the decoder attained in the reconstruction phase is given in Equation (13).
(13)$$z=\sigma (\sum_{u\in R}l^{u}\times {\tilde{W}}^{u}+os)$$

A set of latent feature maps is given as R and bias per input channel is referred to as os
in the reconstruction phase. The acquired deep feature from the autoencoder is given as $Df_{gf}^{acn}$ and they are fed to the concatenation phase.

1DCNN [28]: The acquired data $DA_{wa}^{CH}$ from the channel section is offered as the input to the extraction of deep features to attain the second set of deep features. This 1DCNN model consists of three various parts that are a fully connected layer, a convolutional layer, and a pooling layer. Here 1D signal is passed in the input layer of 1D CNN and a convolution operation is carried out among the input signal and convolution kernels to produce an input feature map. Later, input feature maps are processed over the activation layer for producing an outcome feature map in the convolution layer. Most of the mainstream models are related to two-dimensional convolutions and it has a major difference in the convolution process and they are expressed in Equation (14).
(14)$$q_{1}=conv1y\left(c_{j-1},q_{j-1}\right)+s_{j}$$

Here the term $q_{1}$ is the outcome of $q_{j-1}$ and bias is given as $s_{j}$. The acquired deep feature from 1DCNN is denoted as $Df_{jr}^{1DCNN}$ and they are offered to the concatenation stage. In this phase, two sets of features $\left\{Df_{gf}^{acn},Df_{jr}^{1DCNN}\right\}$ acquired from the autoencoder and 1DCNN are subjected to integration. The achieved integrated features are denoted as $Df_{g}^{cf}=\left\{Df_{gf}^{acn},Df_{jr}^{1DCNN}\right\}$ and are further offered to the accurate selection of the feature phase.

### 5.2. Selection of Accurate Feature

The integrated features $Df_{g}^{cf}$ are offered as the input for the optimal feature selection phase to choose the accurate features by utilizing IDOX for attaining effective features to employ heart disease prediction. The optimal features are utilized to perform efficient prediction of heart disease in the individual and the optimal features are indicated as $Df_{g*}^{of}$ and offered to the heart disease prediction phase. An accurate selection of features based on the developed IDOX is given in Figure 2.

### 5.3. Modified Bi-LSTM-Based Prediction

The acquired optimal features $Df_{g*}^{of}$ are provided as input for the heart disease prediction phase. The parameter such as a suitable number of hidden neuron count in Bi-LSTM is optimized in the range of $\left[2,255\right]$ with the help of improved IDOX. Maximization of accuracy is performed to attain an effective heart disease prediction rate.

A suitable number of hidden neuron counts in BiLSTM is given as $Hd_{i}^{Bilstm}$. Accuracy Acy is a measure of closeness to a particular value as shown in Equation (15).
(15)$$Acy=\frac{\left(hb+hc\right)}{\left(hb+hc+hd+he\right)}$$

Here, the true positive and true negative values are shown as *vb* and *vc*, respectively, and false positive and false negative values are given as vd and ve.

The design of BiLSTM [30] is a result of a Bidirectional RNN (BRNN) network. BiLSTM can fuse several hidden layers with the outcome layer and also resolve different types of issues attained when the allocation is performed in multiple data types in backward as well as the forward direction. Then, the BiLSTM network allocates the forward hidden layer as $x^{⃖}$ and the backward hidden layer as $x^{⃖}$. The forward layers x=1 presented in the output layer E are repeated up to Z and also, for the backward hidden layer x=Z. The outcome is updated with Equations (16)–(18).
(16)$${\vec{x}}_{z}=B\left(J_{m\vec{x}}K_{z}+J_{x^{\to }\vec{x}}{\vec{x}}_{z}+1+p_{x^{⇀}}\right)$$
(17)$${x^{⃖}}_{z}=B\left(J_{mx^{⃖}}K_{z}+J_{x^{⃖}x^{⃖}}x^{⃖}{\;}_{z}+1+p_{x^{⃖}}\right)$$
(18)$$E_{z}=J_{\vec{x}n}{\vec{x}}_{z}+J_{x⃖n}x^{⃖}+c_{n}$$

The outcome vector *E_z_* is achieved with Equation (19).
(19)$$E_{z}=\sigma \left({\vec{x}}_{z},{x^{⃖}}_{z}\right)$$

The term which fuses the output sequence attained in a neuron from the hidden layer is given as σ and also it has four diverse functions that are averaging, summation, multiplication, and concatenation. The secured outcome from BiLSTM is considered a mode trust value. The heart disease prediction model based on M-BiLSTM is represented in Figure 3.

## 6. Results Calculations

### 6.1. Simulation Setup

The simulation performed on the developed prediction of heart disease models in WBAN was developed in Python and also detection model efficacy was computed with several baseline approaches. Effective analysis was performed by utilizing population size as 10 and maximum iteration count as 25. Different classifiers utilized for simulation were Neural Network (NN) [31], K-Nearest Neighbors (KNN) [32], Long Short-Term Memory (LSTM) [33], BiLSTM [34], and Tunicate Swarm-Sail Fish Optimization-Recurrent Neural Network (TS-SFO-RNN) [35] and also heuristic approaches like Grey Wolf Optimization (GWO) [36], Dragonfly Algorithm (DA) [37], DOX [38], Sail Fish Optimization-Tunicate Swarm Algorithm (SFO-TSA) [39] and TS-SFO [40,41].

### 6.2. Efficiency Metrics

The suggested heart disease detection model in WBAN is validated with different quantitative measures [42].

(a) NPV GF is an average of an entire human without affecting disease in a testing stage as given in Equation (20).
(20)$$FG=\frac{hc}{hc+he}$$

(b) Sensitivity SE is the calculations of positive observations which are accurately recognized as given in Equation (21).
(21)$$ES=\frac{hb}{hb+he}$$

(c) FPR EA is a proportion of the negative activities that are inaccurately classified as positive and the average of real negative activities as shown in Equation (22).
(22)$$EA=\frac{vd}{vd+vc}$$

(d) F1-score RD is the proportion of the accuracy rate in the validated test as shown in Equation (23).
(23)$$RD=2\times \frac{2hb}{2hb+hd+he}$$

(e) Specificity FC is the average of negative observations that are accurately recognized as shown in Equation (24).
(24)$$FC=\frac{hc}{hc+hd}$$

(f) MCC WS is a proportion of the number of binary categorizations of testing as shown in Equation (25).
(25)$$WS=\frac{hb\times hc-hd\times he}{\sqrt{\left(hb+hd\right)\left(hb+he\right)\left(hc+hb\right)\left(hc+he\right)}}$$

(g) FNR QD is the average of positives which gives negative test results with the test as shown in Equation (26).
(26)$$QD=\frac{he}{he+hb}$$

(h) Precision EV is the proportion of appropriate features between recovered instances as shown in Equation (27).
(27)$$EV=\frac{hb}{hb+hd}$$

(i) 5-Fold cross-validation (K = 5): In this process, the given data are divided into 5 folds. Here, for the initial iteration, the first fold is fed to the testing stage and the residual data are fed to the training stage. Accordingly, the 2nd fold [43] is used as the testing set while the rest serve as the training set.

### 6.3. K-Fold Analysis of the Suggested Model with Conventional Approaches

Validation on the suggested heart disease prediction model is displayed in Figure 4. Accuracy analysis achieved on the developed IDOX-M-BiLSTM attained 2.83% more effective than GWO-M-BiLSTM, 2.62% enhanced than DA-M-BiLSTM, 1.76% improved than DOX-M-BiLSTM, 4.37% superior to SFO-TSA-M-BiLSTM and 1.24% higher than TS-SFO-RNN models. The given graph results show the success of the designed method regarding K-fold analysis when comparing algorithms. Here, the accuracy rate of the offered method is improved at the 4-fold analysis. The TS-SFO is the second better algorithm. DA is the least algorithm. This degrades the system performance and also generates cross-validation issues. In the same manner, the F1-score of the designed method improves at the 4-fold analysis. While taking the FNR rate, decreases the errors at the 4-fold analysis. Therefore, it is revealed that the better performance of the given designed method attains at the 4-fold analysis. Thus, the offered method achieved an effective heart disease detection rate than baseline approaches.

Various validations performed on the offered method attained a more efficient detection rate than baseline approaches, and they are represented in Figure 5. The F1-score analysis executed on initiated heart disease prediction model achieved effective disease detection rates of 6.15%, 30.18%, 23.2%, 27.77% and 27.76% enhanced than NN, KNN, LSTM, BiLSTM, and TS-SFO-RNN, respectively in 3rd fold. The given graph results show the success of the recommended method regarding K-fold analysis when comparing classifiers. Here, the accuracy rate of the recommended method is improved at the 4-fold analysis. The BiLSTM classifier attains a second better performance. TS-SFO is the least algorithm. Due to this, it generates overfitting issues, and also it has not the ability to resolve optimization problems. In the same manner, the precision of the designed method improves at the 4-fold analysis. While taking the FPR rate, decreases the errors at the 4-fold analysis. Therefore, it is revealed that the better performance of the given designed method attains at the 4-fold analysis. The offered IDXO-based heart disease prediction model based on WBAN achieved a better prediction rate than baseline classifiers in the early stage.

Validation of the offered heart disease identification model based on WBAN with heuristic approaches is displayed in Figure 6. The efficacy analysis performed on initiated heart disease detection model attained an accurate prediction rate than baseline approaches like GWO-M-BiLSTM, DA-M-BiLSTM, DOX-M-BiLSTM, SFO-TSA-M-BiLSTM, and TS-SFO-RNN as 9.71%, 5.26%, 4.0%, 4.3%, and 4.46%, respectively in the learning percentage of 70. The developed model attained an effective high detection rate in accuracy analysis than the conventional approaches. Here, the accuracy rate of the designed method is improved to the 85th learning percentage. The TS-SFO algorithm attains a second better performance. GWO is the least algorithm. Due to this, it has not the ability to resolve optimization problems. Accordingly, the MCC of the designed method improves at the 85th learning percentage. While taking the FDR rate, it decreases the errors at the 85th learning percentage. Therefore, it is revealed that the better performance of the given designed method attains the 85th learning percentage. Thus the developed model based on WBAN archived a more efficient heart disease identification rate than the existing models.

Calculation on the recommended model was weighted up with existing conventional approaches that are NN, KNN, LSTM, BiLSTM, and TS-SFO-RNN and achieved effective heart disease detection rates of 8.9%, 7.7%, 11.4%, 6.5%, and 3.19%, respectively in the learning percentage of 65 and they are displayed in Figure 7. The graph results show the success of the designed method. Here, the accuracy rate of the designed method is improved to the 85th learning percentage. The TS-SFO algorithm attains a second better performance. NN is the least algorithm. Accordingly, the F1-score of the designed method improves at the 85th learning percentage. While taking the FNR rate, decreases the errors at the 85th learning percentage. Therefore, it is revealed that the better performance of the given designed method attains the 85th learning percentage. Thus, the developed IDOX-M-BiLSTM-based heart disease prediction model in WBAN achieved an effective disease prediction rate among individuals in the early stage.

Validation of the suggested heart disease detection in the WBAN model is contrasted over several algorithms displayed in Table 3. The proposed heart disease detection model with IDOX-M-BiLSTM is contrasted and secured superior detection rates of 5.28%, 3.93%, 2.4%, 1.03 and 0.83% over several existing approaches that are GWO-M-BiLSTM, DA-M-BiLSTM, DOX-M-BiLSTM, SFO-TSA-M-BiLSTM, and TS-SFO-RNN, respectively. Thus, the developed heart disease prediction model achieved an effective detection rate in individuals than in the existing models.

### 6.4. Validation of the Suggested Model with Multiple Prediction Algorithms

Computational analyses performed on the developed heart disease prediction model with conventional classifiers are displayed in Table 4. The developed IDOX-M-BiLSTM for heart disease prediction model achieved 3.59%, 3.47%, 6.19%, 2.99%, and 0.54% enhanced prediction rates than NN, KNN, LSTM, BiLSTM, and TS-SFO-RNN, respectively. So, the developed heart disease prediction model achieved an effective prediction rate than the conventional approaches.

### 6.5. Estimation of the Suggested Model with Recent Prediction Algorithms

Estimations on the developed heart disease prediction model with conventional recent prediction algorithms are displayed in Table 5. The precision rate of the developed IDOX-M-BiLSTM for heart disease prediction model achieved 4.23%, 1.18%, and 3.56%, with enhanced prediction rates than PSO-GA, ATSA, and PF-HHO. So, the developed heart disease prediction model achieved an effective prediction rate than the conventional approaches.

## 7. Conclusions

A new prediction of the heart disease model in WBAN has been developed to offer an efficient disease detection rate in the individual by using deep structured architectures. Various data utilized for the analysis such as heartbeat rate, oxygen level, and respiratory rate were achieved from standard sources and offered to the channel selection phase. Then, the attained channels are provided for the deep feature extraction phase. Here, the deep features are attained with the help of 1DCNN and autoencoder further concatenation was performed in the deep feature and fed to the feature selection phase. Later, the feature was selected by utilizing the improved IDOX, and then the acquired features were offered to M-BiLSTM for achieving an efficient heart disease prediction rate by tuning the hidden neuron using improved IDOX and enlarging the accuracy rate. Accuracy analysis achieved on the developed IDOX-M-BiLSTM attained 2.83% more effective than GWO-M-BiLSTM, 2.62% enhanced than DA-M-BiLSTM, 1.76% improved than DOX-M-BiLSTM, 4.37% superior than SFO-TSA-M-BiLSTM and 1.24% higher than TS-SFO-RNN models. Therefore, the offered prediction of the heart disease model secured an efficient disease prediction rate among the individual than the existing approaches. The given designed system will be incorporated with the blood pressure measurement system in upcoming studies. We will investigate more details about the Internet of Medical Things devices which can help humans to self-evaluate their heart conditions. Additionally, it will help to detect diseases in an early stage. We contribute our work to the development of real-time and manual Cardiovascular Disease (CVDs) categorization from Phonocardiogram (PCG) recordings which will create significant real-world impacts in the area of clinical diagnostics.

## Figures and Tables

**Figure 1 biomedicines-11-01167-f001:**
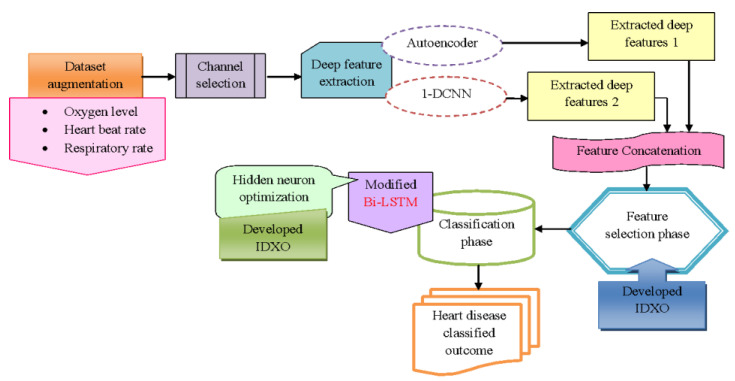
Diagrammatic representation of designed heart disease prediction model in WBAN.

**Figure 2 biomedicines-11-01167-f002:**
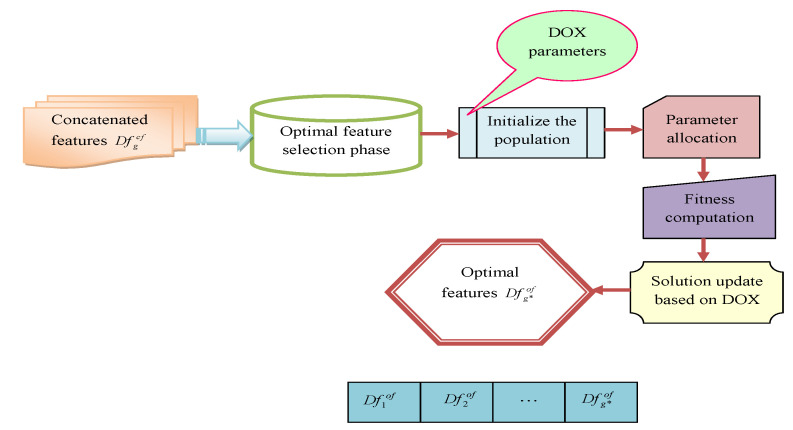
Optimal feature selection in the developed IDOX.

**Figure 3 biomedicines-11-01167-f003:**
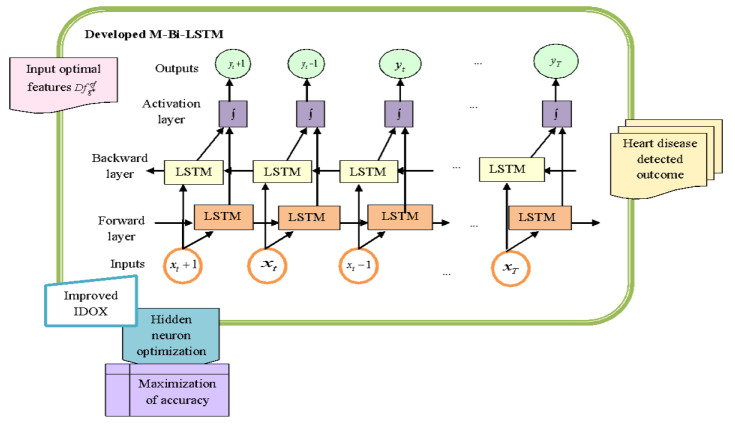
Heart disease prediction model based on the developed M-BiLSTM.

**Figure 4 biomedicines-11-01167-f004:**
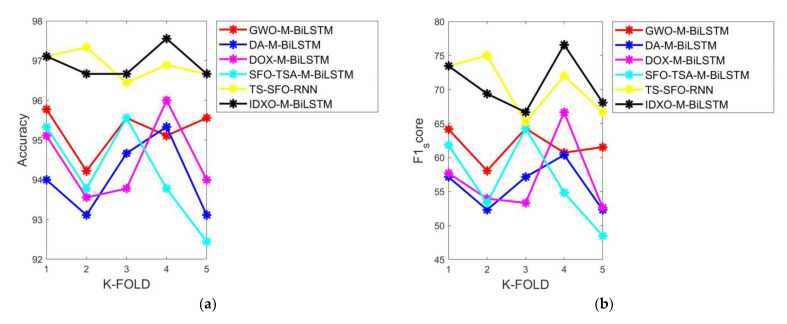
K-fold analysis of the suggested prediction of the heart disease model in WBAN with conventional approaches (GWO-M-BiLSTM [44], DA-M-BiLSTM [45], DOX-M-BiLSTM [46], SFO-TSA-M-BiLSTM [47], TS-SFO-RNN [48]) over (**a**) accuracy, (**b**) F1-score, (**c**) FDR, (**d**) FNR, (**e**) FPR, (**f**) MCC, (**g**) NPV, (**h**) precision, (**i**) sensitivity and (**j**) specificity K-fold analysis of the offered method with conventional prediction algorithms.

**Figure 5 biomedicines-11-01167-f005:**
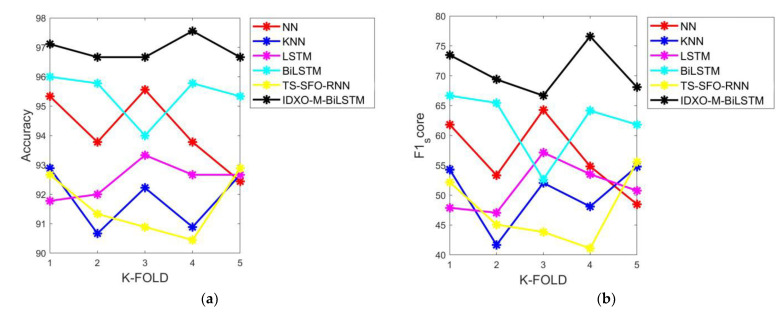
K-fold analysis of the suggested prediction of heart disease model with existing prediction algorithms (NN [49], KNN [50], LSTM [51], BiLSTM [52], TS-SFO-RNN [48]) over (**a**) accuracy, (**b**) F1-score, (**c**) FDR, (**d**) FNR, (**e**) FPR, (**f**) MCC, (**g**) NPV, (**h**) precision, (**i**) sensitivity and (**j**) specificity Analysis of the developed model in WBAN with heuristic approaches.

**Figure 6 biomedicines-11-01167-f006:**
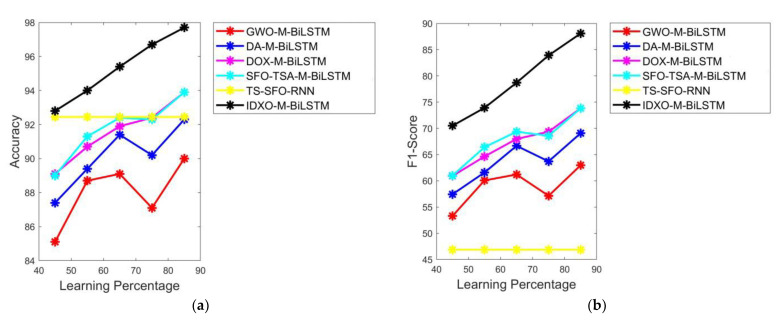
K-fold analysis of the suggested heart disease prediction model with conventional approaches (GWO-M-BiLSTM [44], DA-M-BiLSTM [45], DOX-M-BiLSTM [46], SFO-TSA-M-BiLSTM [47], TS-SFO-RNN [48]) over (**a**) accuracy, (**b**) F1-score, (**c**) FDR, (**d**) FNR, (**e**) FPR, (**f**) MCC, (**g**) NPV, (**h**) precision, (**i**) sensitivity and (**j**) specificity Overall simulation analysis of the suggested model with existing prediction algorithms.

**Figure 7 biomedicines-11-01167-f007:**
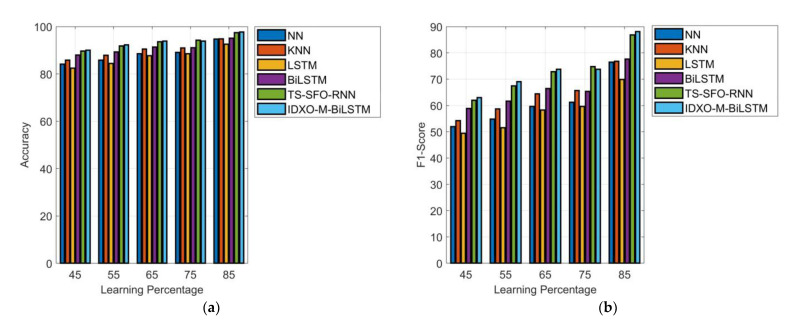
Overall efficacy analysis of the suggested prediction of heart disease model with conventional prediction algorithms (NN [49], KNN [50], LSTM [51], BiLSTM [52], TS-SFO-RNN [48]) over (**a**) accuracy, (**b**) F1-score, (**c**) FDR, (**d**) FNR, (**e**) FPR, (**f**) MCC, (**g**) NPV, (**h**) precision, (**i**) sensitivity and (**j**) specificity validation of the initiated model with a baseline algorithm.

**Table 1 biomedicines-11-01167-t001:** Superiorities and downsides of existing heart disease prediction model in WBAN with deep learning approaches.

Author [Citation]	Framework	Superiorities	Downsides
Guo et al. [1]	RFRF-ILM	It is used to combine random features as well as linear features to offer an accurate disease prediction rate. It effectively minimized the time and cost of the system.	It needs a superior amount of data to perform effective analysis.
Su et al. [2]	STM32	It enhanced the analysis and curve fitting rate rapidly when the input data is provided.	It consumes more time and the system is cost-effective.
Shuvo et al. [3]	CRNN	It offered a high efficacy rate and a non-latency rate.	It can use only limited data for the analysis.
Karhade et al. [4]	TFDDL	It proved superior clarity and maintain simplicity at the time of system analysis.	It required a transfer function to perform the stimulation and they get lagged in time and frequency resolution.
Sonal et al. [5]	Three-tier network	It offered a highly efficient outcome with an enhanced convergence rate by utilizing multiple parameters.	It consumed more time to resolve the occurred issues.
Basheer et al. [6]	HFDT	It effectively removes the redundant structure attained in data and provides an enhanced detection rate.	It required a huge amount of time to connect and they are highly sensitive in a noisy background.
Zang et al. [7]	Markov decision processes	It tunes the energy efficacy attained in the WBAN node as well as the gateway node and also enhanced the throughput.	As it did not consider the power consumption, the system move towards high complexity.
Sarmah. [8]	DLMNN	It utilized a highly secured data transfer process for achieving an optimal outcome.	It did not have the efficacy to monitor heart disease in realtime.

**Table 2 biomedicines-11-01167-t002:** The decision variables and the feasibility constraints for designed IDOX.

Parameters	Values
Number of dingoes	100
Maximum number of iterations	25
Hunting or Scavenger rate	0.5
Maximum number of dingoes that will attack	50
Group attack or persecution rate	0.7
beta1	−2 + 4.094535
beta2	−1 + 2.95045
Number of dingoes that will attack	[2–50]
Group attack	2.2.1
Minimum Bound	−10
Minimum number of dingoes that will attack	2
Maximum Bound	10
dim	30

**Table 3 biomedicines-11-01167-t003:** Validation of the developed IDOX-based heart disease prediction model in WBAN with multiple algorithms.

Measures	GWO-M-BiLSTM [44]	DA-M-BiLSTM [45]	DOX-M-BiLSTM [46]	SFO-TSA-M-BiLSTM [47]	TS-SFO-RNN [48]	IDOX-BiLSTM
Accuracy	92.80	94.00	95.40	96.70	96.89	97.70
Sensitivity	100.00	98.84	98.84	100.00	75.00	98.84
Specificity	92.12	93.55	95.08	96.39	98.12	97.59
Precision	54.43	59.03	65.39	72.27	69.23	79.44
FPR	7.88	6.46	4.92	3.61	1.88	2.41
FNR	0.00	1.16	1.16	0.00	25.00	1.16
NPV	92.12	93.55	95.08	96.39	98.12	97.59
FDR	45.57	40.97	34.62	27.73	30.77	20.56
F1-score	70.49	73.91	78.70	83.90	72.00	88.08
MCC	70.81	73.77	78.29	83.46	70.42	87.46

**Table 4 biomedicines-11-01167-t004:** Validation of developed heart disease prediction model in WBAN with several prediction algorithms.

Measures	NN [49]	KNN [50]	LSTM [51]	BiLSTM [52]	TS-SFO-RNN [48]	IDOX-M-BiLSTM
Accuracy	94.70	94.80	92.60	95.10	96.89	97.70
Sensitivity	100.00	100.00	100.00	98.84	75.00	98.84
Specificity	94.20	94.31	91.90	94.75	98.12	97.59
Precision	61.87	62.32	53.75	63.91	69.23	79.44
FPR	5.80	5.69	8.10	5.25	1.88	2.41
FNR	0.00	0.00	0.00	1.16	25.00	1.16
NPV	94.20	94.31	91.90	94.75	98.12	97.59
FDR	38.13	37.68	46.25	36.09	30.77	20.56
F1-score	76.44	76.79	69.92	77.63	72.00	88.08
MCC	76.34	76.66	70.28	77.27	70.42	87.46

**Table 5 biomedicines-11-01167-t005:** Validation of the developed heart disease prediction model in WBAN with several recent prediction algorithms.

Measures	PSO-GA [53]	ATSA [54]	PF-HHO [55]	IDOX-M-BiLSTM
Accuracy	80.5	82.8	82	97.70
Sensitivity	93.87	94.97	95.83	98.84
Specificity	78.66	81.18	80.41	97.59
Precision	90.60	93.33	91.19	94.44
FPR	21.33	18.81	19.58	2.41
FNR	43.53	27.43	11.62	1.16
NPV	78.66	81.18	80.41	97.59
FDR	69.39	66.66	67.80	20.56
F1-score	86.86	85	86.57	88.08
MCC	49.06	52.02	60.40	87.46

## Data Availability

Data sharing not applicable.

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
