# Peer review of "Intelligent Bi-LSTM with Architecture Optimization for Heart Disease Prediction in WBAN through Optimal Channel Selection and Feature Selection"

_biomedicines, 2023, doi:10.3390/biomedicines11041167_

Round 1

Reviewer 1 Report

Paper is well written. Methods are sound and conclusion well supported by the results.

Additional Comments:

Paper deals with the use of a novel structure of LST; networks for analysis wireless body area networks. Networks are used to predict diseases of the hearth via the analysis and selection of channels and features.

The topic is relevant and it addresses a specific growing area of interest. The paper is sound and it is relevant to the field.

The state of the art is discussed in a sufficient way so the reader is convinced of the innovation and of the novelties of the paper.
Results are presented very well and the number of investigated subject seems to be statistically sufficient.

I would to suggest to discuss about the energy consumption of the WBAN, since this problem is relevant.
Conclusion are consistent and they have address the problem.

Author Response

Paper is well written. Methods are sound and conclusion well supported by the results.

Reply: I thank you for the valuable comments on this paper.

Additional Comments:

Paper deals with the use of a novel structure of LST; networks for analysis wireless body area networks. Networks are used to predict diseases of the hearth via the analysis and selection of channels and features.

The topic is relevant and it addresses a specific growing area of interest. The paper is sound and it is relevant to the field.

The state of the art is discussed in a sufficient way so the reader is convinced of the innovation and of the novelties of the paper.

Results are presented very well and the number of investigated subject seems to be statistically sufficient.

Reply: I thank you for the valuable comments on this paper.

I would to suggest to discuss about the energy consumption of the WBAN, since this problem is relevant.

Reply: The energy consumption of the WBAN has been briefly discussed in the 2nd paragraph of section 1.

Conclusion are consistent and they have address the problem.

Reply: I thank you for the valuable comments on this paper.

Reviewer 2 Report

This work is not enough contribution and innovation. However, the problem statement and motivation could be stronger or more clearly highlighted.

1.      The existing literature should be classified and systematically reviewed, instead of being independently introduced one-by-one.

2.      The abstract is too general and not prepared objectively. It should briefly highlight the paper's novelty as what is the main problem, how has it been resolved and where the novelty lies?

3.      For better readability, the authors may expand the abbreviations at every first occurrence.

4.      The author should provide only relevant information related to this paper and reserve more space for the proposed framework.

5.      However, the author should compare the proposed algorithm with other recent works or provide a discussion. Otherwise, it's hard for the reader to identify the novelty and contribution of this work.

6.      The descriptions given in this proposed scheme are not sufficient that this manuscript only adopted a variety of existing methods to complete the experiment where there are no strong hypothesis and methodical theoretical arguments. Therefore, the reviewer considers that this paper needs more works.

The algorithm presented has not any novelty.

7.      The related works section is very short and no benefits from it. I suggest increasing the number of studies and add a new discussion there to show the advantage.  Following studies can be discussed.

a.      An integrated decision support system for heart failure prediction based on feature transformation using grid of stacked autoencoders

b.      Detection of Cardiovascular Disease Based on PPG Signals Using Machine Learning with Cloud Computing

c.      Exploration of Black Boxes of Supervised Machine Learning Models: A Demonstration on Development of Predictive Heart Risk Score

8.      The manuscript is not well organized. The introduction section must introduce the status and motivation of this work and summarize with a paragraph about this paper.

Author Response

1. The existing literature should be classified and systematically reviewed, instead of being independently introduced one-by-one.

Reply:  Thank you very much for your valuable suggestion; we have incorporated all the changes in the revised manuscript. The related works has now been categorized based on their characteristics in section 2.1.

  1. The abstract is too general and not prepared objectively. It should briefly highlight the paper's novelty as what is the main problem, how has it been resolved and where the novelty lies?

Reply:  Thank you very much for your valuable suggestion; we have incorporated all the changes in the revised manuscript. The novelty of this research work has been highlighted in the 10th line of the abstract section.

  1. For better readability, the authors may expand the abbreviations at every first occurrence.

Reply: Thank you very much for your valuable suggestion; we have incorporated all the changes in the revised manuscript. The abbreviations in the given manuscript have been defined successfully.

  1. The author should provide only relevant information related to this paper and reserve more space for the proposed framework.

Reply:  Thank you very much for your valuable suggestion; we have incorporated the changes in the revised manuscript. As per your suggestion, more details about the proposed work have been provided successfully. Kindly, consider it for paper publication.

5. However, the author should compare the proposed algorithm with other recent works or provide a discussion. Otherwise, it's hard for the reader to identify the novelty and contribution of this work.

Reply:  Thank you very much for your valuable suggestion; we have incorporated all the changes in the revised manuscript. The evaluation of the recent approaches has been added in section 6.5.

6. The descriptions given in this proposed scheme are not sufficient that this manuscript only adopted a variety of existing methods to complete the experiment where there are no strong hypothesis and methodical theoretical arguments. Therefore, the reviewer considers that this paper needs more works.

Reply:  Thank you very much for your valuable suggestion; we have incorporated all the changes in the revised manuscript. As per your suggestion, the manuscript has been improved with more refined points. More details about the proposed work have been provided in the 2nd paragraph of section 3.3 and also the evaluation of the recent approaches has been added in section 6.5. The advantages of the designed method have been added at the end of section 2.1.

The algorithm presented has not any novelty.

Reply: Thank you very much for your valuable suggestion; we have incorporated all the changes in the revised manuscript .The novelty of the algorithm has been highlighted in section 4.1.

7. The related works section is very short and no benefits from it. I suggest increasing the number of studies and add a new discussion there to show the advantage. Following studies can be discussed.

An integrated decision support system for heart failure prediction based on feature transformation using grid of stacked autoencoders

  1. Detection of Cardiovascular Disease Based on PPG Signals Using Machine Learning with Cloud Computing
  2. Exploration of Black Boxes of Supervised Machine Learning Models: A Demonstration on Development of Predictive Heart Risk Score

Reply:  Thank you very much for your valuable suggestion; we have incorporated all the changes in the revised manuscript. The above mentioned recent references has been added in section 2.1 and also the discussion section based on the advantages of the designed method has been added at the end of section 2.1.

  1. The manuscript is not well organized. The introduction section must introduce the status and motivation of this work and summarize with a paragraph about this paper.

Reply: Thank you very much for your valuable suggestion; we have incorporated all the changes in the revised manuscript. The given manuscript has been reorganized and also the motivation of this work has been added in the 7th line of section 3.3.

Reviewer 3 Report

Comment #1: The Introduction and the description of the medical background is difficult to follow and includes many significant errors. There are too many  errors to be listed indivudually

- the authors should have the paper be revised by a clinician knowlegable in heart disease

Comment #2: The authors claim that the described system improves prediction of heart disease. However, it is unclear how Heart Disease was defined and what aspect was better predicted

- please describe in detail

Author Response

Comment #1: The Introduction and the description of the medical background is difficult to follow and includes many significant errors. There are too many  errors to be listed indivudually

- the authors should have the paper be revised by a clinician knowlegable in heart disease

Reply: Thank you very much for your valuable suggestion; we have incorporated all the changes in the revised manuscript. Heart disease and treatment related recent existing works has been added in the 10th line of the 3rd paragraph of section 1. Moreover, it has been cited in the revised manuscript successfully. Also, the related works has been categorized based on their characteristics in section 2.1.

Comment #2: The authors claim that the described system improves prediction of heart disease. However, it is unclear how Heart Disease was defined and what aspect was better predicted

- please describe in detail

Reply: Thank you very much for your valuable suggestion. The aspects for predicting heart disease has been discussed now in section 3.2.

Round 2

Reviewer 2 Report

No further comments.

Reviewer 3 Report

the text describing clinical aspects/background is insufficient and should be revised by a clinician